# Hypoglossal Nerve Stimulation Therapy in a Belgian Cohort of Obstructive Sleep Apnea Patients

**DOI:** 10.3390/life14070788

**Published:** 2024-06-21

**Authors:** Dorine Van Loo, Marijke Dieltjens, Sanne Engelen, Johan Verbraecken, Olivier M. Vanderveken

**Affiliations:** 1Department of ENT, Head and Neck Surgery, Antwerp University Hospital, 2650 Edegem, Belgium; 2Translational Neurosciences, Faculty of Medicine and Health Sciences, University of Antwerp, 2610 Wilrijk, Belgium; 3Multidisciplinary Sleep Disorders Centre, Antwerp University Hospital, 2650 Edegem, Belgium; johan.verbraecken@uza.be; 4Research Group LEMP, Faculty of Medicine and Health Sciences, University of Antwerp, 2610 Wilrijk, Belgium

**Keywords:** obstructive sleep apnea, OSA, sleep apnea, hypoglossal nerve stimulation, HGNS, upper airway stimulation, UAS, treatment outcomes

## Abstract

Hypoglossal nerve stimulation (HGNS) has emerged as a widespread and innovative treatment option for selected moderate-to-severe obstructive sleep apnea (OSA) patients who cannot be treated effectively with traditional treatment modalities. In this observational cohort study, the objective and subjective outcomes of Belgian OSA patients treated with HGNS therapy were analyzed at 6 and 12 months post-implantation. Thirty-nine patients implanted with a respiration-synchronized HGNS device at the Antwerp University Hospital w ere included in this study. Patients underwent baseline in-laboratory polysomnography and a follow-up sleep study 6 and 12 months post-implantation. Questionnaires on patient experience and daytime sleepiness were filled out and data on objective therapy usage were collected. All 39 patients completed the 6-month follow-up and 21 patients (54%) completed the 12-month follow-up. Median AHI decreased from 33.8 [26.1;45.0] to 10.2 [4.8;16.4] at the 6-month follow-up, and to 9.6 [4.1;16.4] at the 12-month follow-up (*p* < 0.001). The surgical success rate, according to the Sher_20_ criteria, was 80% and 76% at the 6- and 12-month follow-ups, respectively. Median ESS improved from 12.0 [7.0;18.0] at baseline to 6.0 [2.5;11.0] at 6 months (*p* < 0.001) and to 6.5 [2.8;11.5] at 12 months (*p* = 0.012). Objective therapy usage was 7.4 [6.6;8.0] and 7.0 [5.9:8.2] h/night at the 6- and 12-month follow-ups, respectively. A high overall clinical effectiveness of HGNS therapy, as shown by a mean disease alleviation of 58%, was demonstrated at 12 months post-implantation. Overall, HGNS therapy using respiration-synchronized neurostimulation of the XII cranial nerve resulted in a significant improvement in both objective and subjective OSA outcomes, with a high level of patient satisfaction and high treatment adherence.

## 1. Introduction

Obstructive sleep apnea (OSA) is a highly prevalent sleep-related breathing disorder in which respiratory events occur during sleep. The main pathophysiological reason for interruptions in the normal breathing pattern during sleep is partial (hypopnea) or complete obstruction (apnea) of the upper airway [1]. These nocturnal respiratory events repeatedly disrupt physiological sleep patterns, leading to intermittent deficiencies in oxygen uptake and nocturnal hypoxemia, thus hindering the recovery of bodily functions during sleep and causing sleep fragmentation [2]. As a result of undiagnosed or untreated OSA, many patients are sleepy throughout the day. This leads to performance degradation and affects the patient’s cognitive performance and quality of life [3,4]. The cumulative effect of this nocturnal oxygen deprivation has a proven negative impact on the patient’s cardiovascular health and causes many other co-morbidities and OSA-related mortality [5,6,7,8,9,10,11].

The traditional non-surgical treatment options for OSA include continuous positive airway pressure (CPAP) and a mandibular advancement device (MAD) [12,13,14,15,16,17]. Surgical treatments for OSA include upper airway surgery, transoral robotic surgery, and maxillomandibular advancement [18,19,20,21]. In addition, treatment options such as positional therapy, myofunctional therapy, and drug therapy are currently available [22,23,24]. For a selected group of patients, respiration-synchronized hypoglossal nerve stimulation (HGNS) or upper airway stimulation (UAS) therapy by the Inspire system (Inspire Medical Systems Inc., Golden Valley, MN, USA) can offer an effective, sustainable solution. Over the past few years, HGNS therapy has emerged as a widespread and innovative treatment option for selected moderate-to-severe OSA patients who cannot be effectively treated with CPAP or MAD. The mechanism of action of the CE-marked and FDA-approved Inspire system relies on the selective stimulation of the hypoglossal nerve in synchronization with the respiratory cycle, causing the tongue muscles that are responsible for forward displacement to contract in order to maintain an open airway [25,26]. A wide range of publications on the outcomes of HGNS therapy is available, including long-term studies that clearly demonstrate the effectiveness and safety of the system [25,27,28,29,30,31,32,33,34,35,36,37,38].

The aim of this study is to report on the results of HGNS therapy using the Inspire system in Belgium in terms of respiratory outcomes, subjective outcome measures, therapy adherence, and patient experience at two different follow-up visits (6 and 12 months post-implantation).

## 2. Materials and Methods

### 2.1. Study Design and Population

This study comprises an analysis of a single-center Belgian cohort of patients who are included in the ADHERE registry. The ADHERE registry is an international multi-center prospective and retrospective observational study that collects real-world clinical outcomes from OSA patients treated with the Inspire system [35]. The registry collects data from different routine clinical visits: pre-implant, implant, post-titration at 6 months post-implantation, and after 12 months. The study flowchart is shown in Figure 1.

Patients were included in this study if they were implanted with the HGNS system (Inspire 4) from Inspire Medical Systems at the Antwerp University Hospital between 2015 and 2023 and if they met the following inclusion criteria: an apnea-hypopnea index (AHI) of between 15 and 65 events/h of sleep, diagnosed by means of a recent (<2 years old) type I full-night attended polysomnography (PSG), a body-mass index (BMI) of <32 kg/m^2^, a combined central and mixed AHI of less than 25%, intolerance to CPAP, and the absence of complete concentric collapse (CCC) at the level of the soft palate, which is determined during drug-induced sleep endoscopy (DISE). These inclusion criteria are aligned with the terms of the FDA approval granted to the Inspire system in 2014, based on findings from earlier studies [25,39,40]. Exclusion criteria were neuromuscular disease, hypoglossal nerve palsy, severe restrictive or obstructive pulmonary disease, moderate-to-severe pulmonary arterial hypertension, severe valvular heart disease, New York Heart Association class III or IV heart failure, recent myocardial infarction or severe cardiac arrhythmias (within the past 6 months), persistent uncontrolled hypertension despite medical treatment, active psychiatric disease, and coexisting non-respiratory sleep disorders that would confound functional sleep assessment [25].

Ethical committee approval from the Antwerp University Hospital was obtained for this study and written informed consent was obtained from all participants.

### 2.2. Hypoglossal Nerve Stimulation Therapy

The CE-marked and FDA-approved Inspire system (Inspire Medical Systems Inc., Golden Valley, MN, USA) consists of a combination of implantable and external devices. There are 3 implantable components: the stimulation lead with the cuff electrode, the implantable pulse generator (IPG), and the breathing sensor. Over the past few years, the implantation technique has evolved from a method with 3 incisions to a modified approach with 2 incisions [41]. The primary alteration entails the positioning of the breathing sensor, which is placed between the internal and external intercostal muscles to detect the breathing pattern. In the 2-incision modified technique, the breathing sensor is now positioned in the second intercostal space instead of the fifth, removing the necessity for the third lower-chest incision. This adjustment allows for the placement of the breathing sensor through the same incision used for the IPG. The stimulation lead consists of a cuff with 3 electrodes, which is placed around the protruding branches of the hypoglossal nerve, including cervical spinal nerve 1 (C1), using a selective nerve integrity monitoring (NIM) system [42]. The IPG, which contains the battery and synchronizes the stimulation with the breathing, is implanted in the right ipsilateral mid-infraclavicular region. The implantation of the HGNS system is performed according to previously published protocols [42,43,44].

The external components of the Inspire system include the patient remote and physician programmer [44]. The patient remote is used by the patient to turn the therapy on and off; it also allows the patient to increase and/or decrease the stimulation amplitude within a pre-defined range. Conversely, the physician programmer, which consists of a tablet computer and a telemetry unit, is used to obtain wireless communication with the IPG via short-range radiofrequency telemetry. This telemetry communication allows the physician to conduct noninvasive assessments of IPG status, including battery levels and therapy usage. Moreover, it allows adjustments of the stimulation and sensing parameters and the monitoring of respiratory waveforms. The telemetry unit operates via a wall outlet connection and communicates wirelessly via Bluetooth with the physician programming tablet [44].

The post-implant care pathway starts with the device activation, which is typically performed around four weeks post-implantation during a daytime clinical visit. During this visit, the patient is educated on how to use the therapy and the patient remote. Following activation, there is a period of therapy adaptation, as well as close monitoring. During this period, patients gradually increase the stimulation strength by using the remote [44]. Around three months post-activation, a fine-tuning sleep study is performed. Depending on the outcome of the sleep study, the patients are directed onto either a green or a yellow pathway. The green pathway signifies achieving a 50% reduction in AHI and an on-therapy AHI of <15 events/h, plus a therapy usage of more than 4 h per night. Failure to meet one of these criteria or both of them will direct patients to the yellow pathway, necessitating further therapy optimization [45].

### 2.3. Outcome Measures

All patients underwent a baseline type I full-night attended PSG. Subsequently, at the 6-month post-implantation follow-up, either the treatment AHI from an in-laboratory titration PSG or a full-night AHI from a home sleep test (HST) was collected. The treatment AHI is defined as the AHI measured under the therapeutic settings for this period during the night. At the 12-month follow-up, AHI was collected from either a type 1 full-night attended PSG or HST.

Scoring of the respiratory events was performed according to the criteria outlined by the American Academy of Sleep Medicine in 2014. An apnea was defined as a minimum of 90% decrease in airflow from baseline for at least 10 s, whereas a hypopnea was defined as a reduction in airflow of at least 30% from baseline for more than 10 s, associated with either an arousal or a drop in oxygen saturation level exceeding 3%. The oxygen desaturation index (ODI) was defined as the mean number of oxygen desaturations of 3% or more per hour.

Surgical success was defined according to two definitions of the Sher criteria (Sher_20_ and Sher_15_): a reduction in baseline AHI of 50% or more and a postoperative AHI of less than 15 (Sher_15_) or 20 (Sher_20_) events/h [46]. Objective therapy usage was retrieved from data stored in the IPG at both the 6- and 12-month follow-ups.

The mean disease alleviation (MDA) is a measure of overall clinical effectiveness and is defined by the product of therapeutic efficacy and adjusted adherence, divided by 100. Therapeutic efficacy is defined as the delta AHI (the reduction in AHI from baseline to 12 months post-implantation, expressed as the percentage of the baseline AHI). The adjusted adherence is calculated as the objective hours of use, corrected for total sleep time [47].

The Epworth sleepiness scale (ESS) questionnaire, a measure of the extent of daytime sleepiness, was completed by the patients at baseline and at 6 and 12 months post-implantation [48]. A questionnaire on the patient’s experience was filled out during both follow-up visits. This questionnaire includes four questions on patient satisfaction: (1) How does HGNS therapy compare against your previous experience with CPAP? (2) What is the likelihood of choosing HGNS therapy again? (3) What is the likelihood of recommending HGNS therapy to friends/family? (4) Overall, how satisfied are you with HGNS therapy?

### 2.4. Statistical Analysis

Statistical analysis was performed using SPSS (SPSS V.27.0, SPSS Inc, Chicago, IL, USA). Normally distributed data are presented as mean ± standard deviation, while non-normally distributed data are presented as median [Q1;Q3]. Changes from the baseline to subsequent follow-up visits were tested using a paired *t*-test for normally distributed parameters and a paired Wilcoxon signed-rank test for non-normally distributed parameters. A *p*-value of <0.05 is considered to be statistically significant.

## 3. Results

### 3.1. Study Population

A total of 39 patients implanted with the Inspire device at the Antwerp University Hospital were included in this study. All 39 patients completed 6 months of follow-up and 21 patients (54%) completed 12 months of follow-up.

An overview of the baseline characteristics is shown in Table 1. The study population is primarily male (79%), middle-aged, and overweight. Mean systolic and diastolic blood pressure was 131.6 ± 16.3 mmHg and 82.7 ± 11.1 mmHg, respectively. Median baseline AHI was 33.8 [26.1;45.0] events/h with a median ESS score of 11.0 [7.0;18.0]. Median ODI ≥ 3% was 24.9 [12.9;31.2] events/h. Twenty-seven patients (69%) had a history of severe OSA (AHI > 30 events/h). Ten patients (25%) had previously undergone palatal surgery.

Twenty-five patients (64%) were implanted using the 3-incision technique and 14 patients (36%) were implanted using the 2-incision modified technique.

### 3.2. Outcome Measures at 6 Months Post-Implantation

An overview of both the objective and subjective treatment outcomes in all 39 patients at 6 months post-implantation is shown in Table 2. There was no significant change in mean BMI from baseline to the 6-month follow-up (27.5 ± 2.8 kg/m^2^ to 27.3 ± 2.8 kg/m^2^; *p* > 0.05). Median AHI decreased from 33.8 [26.1;45.0] events/h to 10.2 [4.8;16.4] events/h at 6 months (*p* < 0.001; Figure 2). The absolute AHI reduction from baseline to 6 months was −23.6 [−32.7;−14.9] corresponding to an overall reduction of 70% with HGNS therapy compared to baseline. ODI ≥ 3% significantly decreased from 25.5 [12.2;31.5] events/h at baseline to 7.9 [5.2;16.7] events/h at 6 months post-implantation (*p* < 0.001), representing a 69 percent median reduction in ODI by HGNS.

The surgical success rate according to the Sher_20_ criteria was 80% at the 6-month follow-up. Sher_15_ criteria were met by 69% of the patients.

Objective therapy usage was 7.4 [6.6;8.0] h/night. Furthermore, 97% of the patients exhibited a therapy usage of more than 4 h per night.

Median ESS improved from 12.0 [7.0;18.0] at baseline to 6.0 [2.5;11.0] at 6 months (*p* < 0.001). The percentage of patients with an ESS score below 10 increased from 40% at baseline to 66.7% at the 6-month follow-up. The response rate for filling out the questionnaire on patient experience was 18/39 (46%) at the 6-month follow-up. Of these 18 patients, all said that they preferred HGNS over CPAP therapy and would choose HGNS therapy again. Similarly, all 18 patients (100%) would recommend HGNS therapy to friends/family and were satisfied with the Inspire 4 HGNS therapy overall.

### 3.3. Outcome Measures at 12 Months Post-Implantation

An overview of both the objective and subjective treatment outcomes in 21 patients at 12 months post-implantation is shown in Table 3. There was no significant change in mean BMI from baseline to the 12-month follow-up (27.5 ± 2.8 kg/m^2^ to 27.1 ± 3.5 kg/m^2^; *p* > 0.05). Median AHI decreased from 37.0 [27.6;43.1] events/h to 10.7 [4.4;15.2] events/h at 6 months, and to 9.6 [4.1;16.4] events/h at 12 months (*p* < 0.001; Figure 2). The absolute AHI reduction from baseline to the 12-month follow-up was −24.8 [−32.2;−16.5]. ODI ≥ 3% significantly decreased from 28.1 [19.4;33.8] events/h at baseline to 9.4 [4.8;16.7] events/h at 6 months, and to 15.9 [3.9;21.7] at 12 months (*p* < 0.001).

The surgical success rate according to the Sher_20_ criteria was 76% at the 12-month follow-up. Sher_15_ criteria were met by 57% of the patients 12 months after implantation.

Objective therapy usage was 7.0 [5.9;8.2] h/night, with a therapy usage of more than 4 h per night in 94% of the patients.

Median ESS improved from 12.0 [6.8;15.8] at baseline to 5.0 [2.0;10.5] at 6 months, and to 6.5 [2.8;11.5] at 12 months (*p* = 0.012; Figure 3). The percentage of patients with an ESS score below 10 increased from 40% at baseline to 75% at the 12-month follow-up. The response rate for the questionnaire on patient experience was 18/21 (86%) for the 12-month follow-up. Of these 18 patients, all preferred HGNS over CPAP therapy. Furthermore, 92% of the patients would choose HGNS therapy again and would recommend HGNS therapy to friends/family. Overall, 92% of the patients were satisfied with HGNS therapy at the 12-month follow-up.

Therapeutic efficacy, expressed as the delta AHI percentage, was 64.4% at the 12-month follow-up. Adjusted adherence, expressed as the objective therapy usage corrected for total sleep time, was 90.1%. The MDA as a measure of overall clinical effectiveness was 58% at the 12-month follow-up (Figure 4).

## 4. Discussion

This observational study showed that respiration-synchronized HGNS therapy using the Inspire 4 implant significantly reduced OSA severity and was accompanied by improved daytime sleepiness, a high patient satisfaction score, and high therapy usage of 7.4 [6.6;8.0] and 7.0 [5.9;8.2] h/night at the 6- and 12-month follow-ups, respectively. The surgical success rate, according to the Sher_20_ criteria, was 80% and 76% at the 6- and 12-month follow-ups, respectively. Our results showed that HGNS therapy results in a durable response, as the significant improvements achieved in both objective and subjective therapy outcomes are maintained at 12 months post-implantation. To our knowledge, we are only the third study to investigate the MDA for HGNS therapy [49,50]. We have demonstrated a high overall clinical effectiveness of HGNS therapy, as shown by an MDA of 58% at 12 months post-implantation.

After the completion of the initial STAR trial, which led to FDA approval of the Inspire system in 2014, a large multi-center observational registry, the ADHERE registry, was initiated to collect data from patients treated with HGNS therapy within routine clinical care. Multiple intermediate analyses have been performed on patients included in this ADHERE registry [32,33,34,35,50]. Thaler et al. showed a significant AHI reduction from 32.8 (interquartile range (IQR), 23.6–45.0) at baseline to 6.3 (IQR, 2–14.8) at the 6-month follow-up, and to 9.5 (IQR, 4.0–18.5) at 12 months post-implantation. Sher_20_ criteria were met by 83% (*n* = 485/582) and 69% (*n* = 265/381) of the participants after 6 and 12 months, respectively [34]. The most recent publication regarding the ADHERE registry is by Bosschieter et al., which reported a significant AHI reduction from 33.0 events/h at baseline to 7.8 events/h at 6 months, and to 10.2 events/h at 12 months. Out of 593 patients, 66.6% met the Sher_20_ criteria at 12 months post-implantation [50]. In addition to the results from the ADHERE registry, multiple other studies investigating HGNS therapy have been published over the years. In a multicenter German post-market study, Steffen et al. reported a significant AHI reduction from 28.6 [21.6;40.1] at baseline to 8.3 [5.2;17.3] at 6 months, and to 9.5 [4.6;18.6] at 12 months. Sher_20_ criteria were met by 73% of the participants at 12 months post-implantation [36]. In our analysis of a Belgian cohort of patients treated with HGNS therapy, the objective outcomes are comparable to earlier findings from the multi-center ADHERE dataset and German post-market study. The surgical success rate at 12 months post-implantation is higher in our cohort compared to the results from the multi-center ADHERE dataset. 

The median objective therapy usage was 5.7 (IQR, 4.1–7.1; mean, 5.6  ±  2.1) h per night at 12 months post-implantation, as reported by Thaler et al. [34]. In the German post-market study, an objective therapy usage of 5.6 ± 2.1 h per night was reported at 12 months post-implantation [36]. The therapy adherence in our cohort is higher in comparison to those findings, at both the 6- and 12-month follow-ups. Until recently, there was no reimbursement for HGNS therapy available in Belgium. The limited number of patients that could be treated with this innovative therapy could have resulted in higher patient motivation and more consistent use of the therapy. In addition, we have implemented a well-structured postoperative care pathway, which is key to ensuring long-term therapy success, including maintaining high therapy adherence.

The MDA, which considers both AHI reduction and therapy adherence, offers a more holistic understanding of treatment effectiveness compared to focusing solely on separate parameters. Moreover, it offers a valuable tool that enables the comparison of different treatment modalities. Heiser et al. demonstrated an MDA of 59% in patients treated with HGNS, compared to 51% among patients receiving CPAP therapy at a 12-month follow-up [49]. Our study showed an MDA of 58% at 12 months post-implantation, which is in line with those findings showing a higher overall clinical effectiveness for HGNS compared to CPAP, on average.

In terms of subjective outcomes, Thaler et al. reported reduced daytime sleepiness, with an ESS score of 11.0 (IQR, 7.0–16.0) at baseline that decreased to 7.0 (IQR, 4.0–11.0) at 6 months and to 6.0 (IQR, 3.0–10.0) at 12 months [34]. Bosschieter et al. described a similar reduction in ESS from 11.0 at baseline (*n* = 1712) to 7.0 at 6 months (*n* = 1528) and to 6.0 at 12 months (*n* = 994) [50]. Steffen et al. demonstrated a reduction in ESS from 13 [9.5;17] at baseline to 6.0 [4;10] at 6 months, and to 6.5 [3;10] at 12 months [36]. In our cohort, the improvement in ESS is similar compared to the studies mentioned above.

Recently, an expansion of the indications for HGNS therapy (Inspire system) has been approved by the FDA based on current findings, including those from the ADHERE registry [35,50]. Bosschieter et al. performed an analysis on the ADHERE dataset that investigated the effect of HGNS therapy in 5 subgroups that were stratified by preoperative OSA severity, including a group with a baseline AHI of >65 events/h. This analysis demonstrated no significant differences between these subgroups regarding treatment success and improvement in self-reported outcomes [50]. Furthermore, Suurna et al. showed a similar AHI reduction and ESS improvement among patients with a baseline BMI of ≤32 kg/m^2^ and patients with a baseline BMI of between 32 and 35 kg/m^2^ [35]. Based on the findings described above, the FDA approved an increase in the upper limit of the AHI from 65 events/h to 100 events/h and the BMI limit has been raised from 32 kg/m^2^ to 40 kg/m^2^.

Our analysis included 25 patients (64%) who were implanted using the 3-incision technique and 14 patients (36%) who underwent implantation with the 2-incision modified technique. In the 2-incision modified technique, the placement of the breathing sensor has moved from the fifth to the second intercostal space, eliminating the third lower-chest incision. As a result, the sensor can be placed using the same incision as is used for the IPG and only one tunnel from the stimulation lead to the IPG is required. The study by Kent. et al. showed that this modified 2-incision technique had a statistically non-inferior therapeutic efficacy compared to a cohort previously implanted via the 3-incision technique, which was 1:1 propensity score-matched for a non-inferiority analysis of postoperative outcomes [41].

One of the limitations of this study is the limited sample size, given that until recently, no reimbursement was available in Belgium. Twenty-one patients (54%) completed the 12-month follow-up. For 10 patients from whom data collection was performed retrospectively, no 12-month follow-up data were available. For the other 8 patients, the 12-month follow-up could not be completed due to the COVID-19 pandemic, other medical issues prevented them from coming to the hospital for a PSG or because they were lost to follow-up.

Another limitation is that data collection for AHI at the 6-month follow-up is performed using different methods, including in-laboratory PSG, HST, or titration PSG. The titration or treatment AHI is defined as the AHI measured under the therapeutic settings for this period during the night and, thus, does not always represent the value for the entire night. A recent study by Kent et al. showed that a titration PSG may not be necessary for all patients. This study demonstrated equivalent improvements in both objective and subjective OSA outcomes in 60 patients who were randomly assigned to either a 3-month post-activation in-laboratory titration PSG or an efficacy HST [51].

Despite its limitations, this study succeeded in demonstrating sustained high overall clinical effectiveness among our Belgian cohort of OSA patients, compared to the literature. This highlights the importance of implementing a structured post-implant care pathway and closely monitoring patients to guarantee long-term treatment success. Notably, this pre-reimbursement study contributed to the reimbursement approval of HGNS therapy in Belgium in 2023.

## 5. Conclusions

This analysis of a Belgian cohort of patients treated with respiration-synchronized HGNS therapy shows significant improvements in both objective and subjective OSA outcomes, with a high patient satisfaction score and high therapy adherence. A significant reduction in OSA severity was demonstrated, accompanied by improved daytime sleepiness at both 6 and 12 months post-implantation. Moreover, a high overall clinical effectiveness of HGNS therapy was noted, as shown by an MDA of 58% at 12 months post-implantation.

## Figures and Tables

**Figure 1 life-14-00788-f001:**
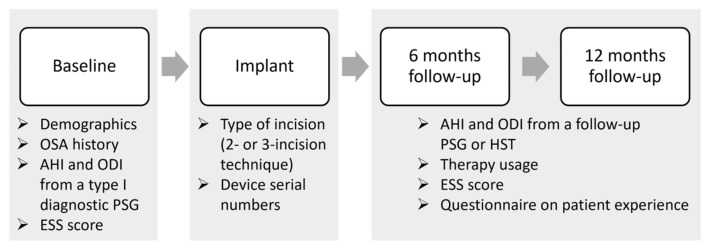
Data collection flowchart. The study collected data from routine clinical visits, including pre-implant, implant, and post-titration at 6 months post-implantation, and after 12 months. Demographics, OSA history, polysomnographic data including apnea-hypopnea index (AHI) and oxygen desaturation index (ODI), and the Epworth sleepiness scale (ESS) score were collected at the baseline or pre-implant visit. The type of incision (2-incision or 3-incision technique) and device serial numbers were retrieved during the implant visit. At both the 6- and 12-month follow-ups, the AHI and ODI were collected from a follow-up sleep study (PSG of HST), while therapy usage was retrieved from data stored in the implantable pulse generator, and questionnaires on patient experience and the ESS were filled out by the patients. Abbreviations: PSG = polysomnography; AHI = apnea-hypopnea index; ODI = oxygen desaturation index; ESS = Epworth sleepiness scale; HST = home sleep test.

**Figure 2 life-14-00788-f002:**
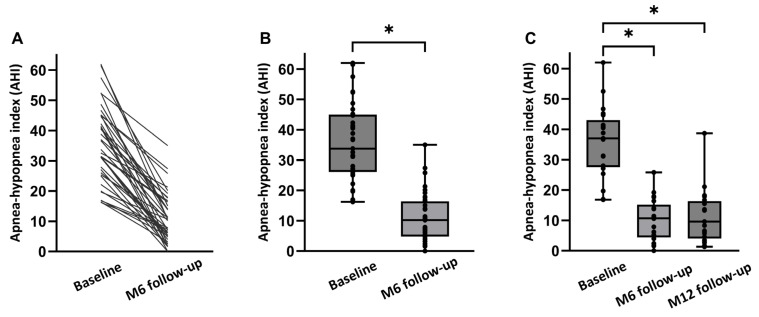
AHI at baseline and at follow-up visits. (**A**) Data in 39 individual patients. (**B**) Median AHI (baseline, 6 months follow-up) (*p* < 0.001). (**C**) Median AHI (baseline, at 6 months (M6), and at 12 months (M12) for the full dataset (*n* = 21)). Individual AHI values of the patients are shown as dots. * *p* < 0.05.

**Figure 3 life-14-00788-f003:**
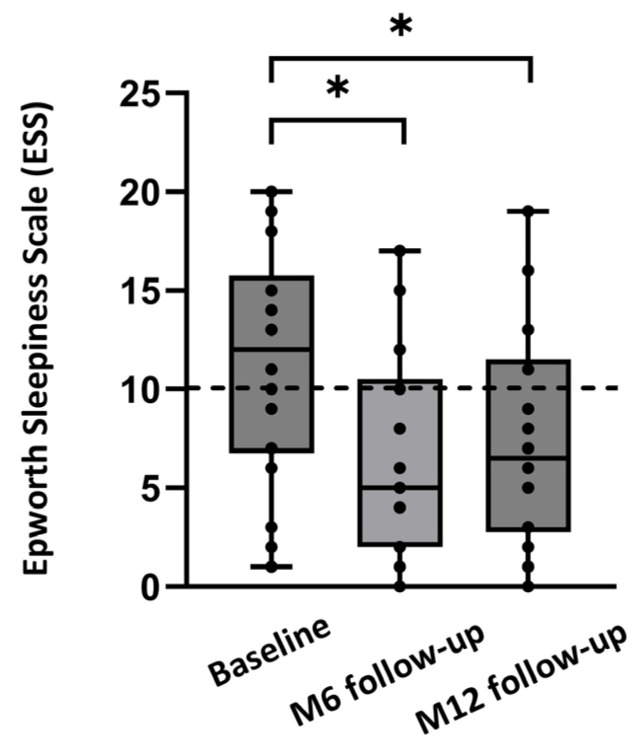
Median ESS at baseline and after 6 months (M6) and 12 months (M12) of follow-up, for the full dataset (*n* = 18). Individual ESS values of the patients are shown as dots. Values below 10 are considered to be normal (dashed line). * *p* < 0.05.

**Figure 4 life-14-00788-f004:**
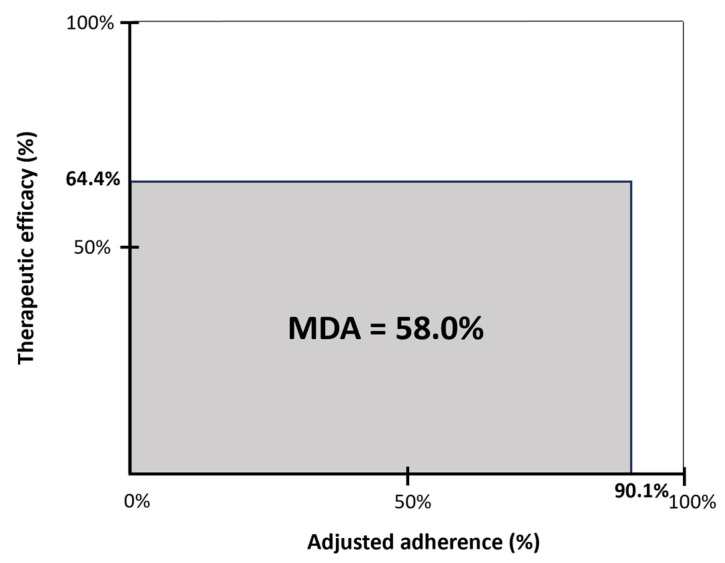
Mean disease alleviation (MDA) for HGNS therapy at 12 months post-implantation. The MDA is a measure of overall clinical effectiveness and is defined by the product of therapeutic efficacy and adjusted adherence, divided by 100. Therapeutic efficacy is defined as the delta AHI (reduction in AHI from baseline to 12 months post-implantation, expressed as the percentage of the baseline AHI). The adjusted adherence was calculated as the objective hours of use, corrected for total sleep time (*n* = 17).

**Table 1 life-14-00788-t001:** Baseline characteristics (*n* = 39).

Characteristic	Value
Age (years)	55.0 ± 11.9
Gender (% male)	79%
BMI (kg/m^2^)	27.5 ± 2.8
Systolic blood pressure (mmHg)	131.6 ± 16.3
Diastolic blood pressure (mmHg)	82.7 ± 11.1
AHI (events/h)	33.8 [26.1;45.0]
ODI ≥ 3% (events/h)	24.9 [12.9;31.2]
ESS score	11.0 [7.0;18.0]

Values are presented as mean ± standard deviation or median [Quartile 1; Quartile 3] unless otherwise specified. Abbreviations: BMI = body mass index; AHI = apnea-hypopnea index; ODI = oxygen desaturation index; ESS = Epworth sleepiness scale.

**Table 2 life-14-00788-t002:** Objective and subjective outcome measures at the 6-month follow-up.

Parameter	Baseline	Month 6	Change	*p*-Value
BMI (kg/m^2^)	27.5 ± 2.8	27.3 ± 2.8	-	>0.05
AHI (events/h)	33.8 [26.1;45.0]	10.2 [4.8;16.4]	−23.6 [−32.7;−14.9]	<0.001
ODI ≥ 3% (events/h)	25.5 [12.2;31.5]	7.9 [5.2;16.7]	−14.0 [−20.8;−3.9]	<0.001
ESS	12.0 [7.0;18.0]	6.0 [2.5;11.0]	−5.0 [−8.5;−1.0]	<0.001
Therapy usage (h/night)	-	7.4 [6.6;8.0]	-	-

Values are presented as mean ± standard deviation or median [Quartile 1; Quartile 3] unless otherwise specified. Abbreviations: BMI = body mass index; AHI = apnea-hypopnea index; ODI = oxygen desaturation index; ESS = Epworth sleepiness scale.

**Table 3 life-14-00788-t003:** Objective and subjective outcome measures at the 12-month follow-up.

Parameter	Baseline	Month 12	Change	*p*-Value
BMI (kg/m^2^)	27.5 ± 2.8	27.1 ± 3.5	-	>0.05
AHI (events/h)	37.0 [27.6;43.1]	9.6 [4.1;16.4]	−24.8 [−32.2;−16.5]	<0.001
ODI ≥ 3% (events/h)	28.1 [19.4;33.8]	15.9 [3.9;21.7]	−11.7 [−18.6;−3.1]	<0.001
ESS	12.0 [6.8;15.8]	6.5 [2.8;11.5]	−3.0 [−9.3;−1.0]	0.012
Therapy usage (h/night)	-	7.0 [5.9;8.2]	-	-

Values are presented as mean ± standard deviation or median [Quartile 1; Quartile 3], unless otherwise specified. Abbreviations: BMI = body mass index; AHI = apnea-hypopnea index; ODI = oxygen desaturation index; ESS = Epworth sleepiness scale.

## Data Availability

The data presented in this study are available on request from the Antwerp University Hospital, Belgium.

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
