# Peer review of "Hypoglossal Nerve Stimulation Therapy in a Belgian Cohort of Obstructive Sleep Apnea Patients"

_life, 2024, doi:10.3390/life14070788_

Round 1

Reviewer 1 Report

Comments and Suggestions for Authors

The authors present an interesting study to investigate the outcome of HGNS therapy using the Inspire system in Belgium in terms of respiratory outcomes, subjective outcome measures, therapy adherence and patient experience at two different follow-up visits (6- and 12-months 59 post-implantation). It is a very interesting study, however several minor concerns needs to be addressed.

1. the Abstract needs to clarify the aim

2. Justification of this study and how this study adds to the current knowledge

3. How many surgeons/sleep specialists were involved in HGNS insertion, and what years of experience?

4. The methodology needs to be more explicit

5. The discussion session-please include a literature search and compare the outcome from this study and other studies using INSPIRE

Reviewer 2 Report

Comments and Suggestions for Authors

1.      The authors conducted a research study on using the Inspire Upper Airway Stimulation System as a treatment for selected patients with obstructive sleep apnea in a single-center Belgian cohort. The study is part of and follows Inspire Medical’s ADHERE Registry protocol, a multi-center observational registry integrating results from different regions worldwide.

2.    The main question addressed by the research is whether the Inspire Device improves objective and subjective parameters of sleep apnea and to what extent. It would be helpful to identify studies from other centers published in the literature and highlight the new contributions your study brings. For example, some studies have tested the technique on different weight groups and at different AHI intervals at diagnosis, potentially extending the indications to new categories of patients. What new aspects does your study bring? Are there any particularities you observed following your research that might add value to the existing literature?

3.  In the introduction section, the authors mention that treatment options for OSA have so far included only CPAP and mandibular advancement devices, without mentioning surgical interventions at pharyngeal and palatal levels or other treatment options.

4. I have some observations for the methodology section. It would be beneficial to include detailed exclusion criteria for the study participants. Also, while the inclusion criteria are well-detailed and align with the first recommendations for the Inspire UAS system from 2014, it would be helpful to explain the rationale behind utilizing these specific criteria for inclusion. Additionally, it's important to specify the recruitment period for participants, as this detail is currently missing. Lastly, the study is described as both prospective and retrospective, while the ADHERE registry is a prospective multicentre study that collects real-world evidence data.

5.     The conclusion section is clear and concise. I suggest that the statements be accompanied by statistical results to better highlight your findings.

6.       The references list is comprehensive and well-chosen. I recommend adding up-to-date information about the ADHERE register situation and how the indications have been expanded and modified following recent studies. 
